# The Characteristics of Cognitive Proficiency in Patients with Acute Neuromyelitis Optica Spectrum Disease and its Correlation with Serum Aquaporin-4 Antibody Titer

**DOI:** 10.3390/brainsci13010090

**Published:** 2023-01-02

**Authors:** Yan Miao, Xiaoling Zhong, Shuangshuang Jia, Yang Bian, Jinming Han, Feng Qiu

**Affiliations:** 1Department of Neurology, The Third Xiangya Hospital, Central South University, Changsha 410008, China; 2Department of Neurology, School of Medicine, South China University of Technology, Guangzhou 510006, China; 3Department of Neurology, The Sixth Medical Center of Chinese PLA General Hospital of Beijing, Beijing 100048, China; 4Department of Neurology, Naval Clinical College, Anhui Medical University, Hefei 230032, China; 5Department of Neurology, Xuanwu Hospital, Capital Medical University, Beijing 100053, China; 6Senior Department of Neurology, The First Medical Center of PLA General Hospital, Beijing 100853, China

**Keywords:** neuromyelitis optica spectrum disease, cognitive dysfunction, aquaporin-4 antibody, correlation, characteristics, titers

## Abstract

Objective: To explore the characteristics and dynamic evolution of cognitive impairment in patients with neuromyelitis optica spectrum disorder (NMOSD). Methods: Twenty-five patients with acute NMOSD and 30 age-matched healthy individuals were consecutively recruited in this study. The Montreal Cognitive Assessment (MoCA), Chinese Version of Rey Auditory Vocabulary Learning Test (CRAVLT), Verbal Fluency Test (VFT), Digital Span Test (DST), Paced Auditory Serial Addition Task 3/2s version (PASAT-3/2), Rey–Osterrieth Complex Figure Test (ROCF) and Stroop Color and Word Test (CWT) were used to evaluate cognitive function. The correlations between cognitive function and serum aquaporin-4 (AQP-4) antibody titer were analyzed. Results: Sixty-four percent of patients with acute NMOSD had cognitive dysfunction. MoCA (*p* < 0.001), CRAVLT-N7 (*p* = 0.004), CRAVLT-N8 (*p* = 0.011), ROCF-C (*p* = 0.005), ROCF-R (*p* < 0.001), PASAT-3 (*p* = 0.013), PASAT-2 (*p* = 0.001) and CWT-A (*p* = 0.017) were significantly worse in patients with acute NMOSD than those in control group. During follow-up visits, significant differences of serum AQP-4 antibody titers were still noted in NMOSD patients (*p* < 0.001), while no significant differences were found by MoCA. Conclusion: A high number of patients with acute NMOSD suffer from cognitive dysfunction. Serum AQP-4 antibody titers can decrease during disease remission, while obvious cognitive decline in these patients still exists.

## 1. Introduction

Neuromyelitis optica spectrum disorders (NMOSD) are inflammatory disorders of the central nervous system (CNS), mainly involving the optic nerve and spinal cord [1,2,3]. AQP-4 is a transmembrane protein, which is expressed on the terminal foot of astrocytes around blood vessels. AQP-4 regulates the water movement between blood, brain and cerebrospinal fluid, serving as a key part of brain clearance system, brain interstitial fluid and metabolic wastes (such as soluble amyloid protein-β) [1,4]. Metabolic wastes could be drained through the AQP-4-dependent trans-astrocytic glymphatic system or perivascular drainage, and cleared through the overall flow of interstitial fluid [2,4]. It has been demonstrated that the fluid dynamics problem in the brain clearing system exists in many cognitive disorders, and the mechanism of AQP-4 in cognitive disorders has also received extensive attention [5,6,7].

Aquaporin-4 IgG (AQP-4-IgG) has been identified as a specific marker for NMOSD and over 70% of NMOSD patients are positive for serum AQP-4 IgG antibody [8,9,10,11]. In NMOSD, AQP-4-IgG binds to AQP-4, activating the complement-forming membrane attack complex and leading to astrocyte damage. It has been documented that 26% ~67% of patients with NMOSD are accompanied by cognitive impairment during different disease stages [12,13,14,15]. Compared with AQP-4-IgG-negative NMOSD, cognitive function was significantly poor in patients with AQP-4-IgG-positive NMOSD [16]. Neuropathological studies suggested that numbers of neurons in the para-layer and basal layer of the cerebral cortex were obviously damaged in patients with NMOSD [17]. In fact, there was a permanent interaction between activated astrocytes in the first layer and AQP-4-IgG, with non-lytic reactions occurring in AQP-4-negative astrocytes [17]. Due to the lack of studies regarding the relationship between NMOSD and AQP-4 antibody titers, the aim of this study was to explore the characteristics of cognitive impairment of NMOSD during the acute disease stage and ascertain whether there is a potential correlation between cognitive impairment and serum AQP-4 antibody titer. Dynamic evolution of cognitive impairment and serum AQP-4 antibody titers during acute and remissive stages will also be explored.

## 2. Materials and Methods

### 2.1. Patients

Patients with NMOSD were consecutively recruited in the Department of Neurology, General Hospital of PLA from August 2019 to March 2021. This study was approved by the ethics committees of Chinese PLA General Hospital and was conducted in accordance with the Declaration of Helsinki. Written informed consent was obtained from all subjects.

### 2.2. Selection Criteria of Subjects

The inclusion criteria of this study: patients with NMOSD met the latest diagnostic guidelines [18] and did not receive corticosteroid or immunosuppressant therapy after disease onset. This study excluded subjects as follows: (1) Alzheimer’s disease, frontotemporal dementia, vascular dementia and dementia with Lewy bodies. (2) A history of acute cerebrovascular disease within 3 months. (3) Active epilepsy. (4) A history of primary mental disorders. (5) Inability to finish brain magnetic resonance imaging (MRI) examinations. (6) Supratentorial lesions. (7) Liver, kidney or heart dysfunction or multiple organ dysfunction. (8) Other factors that interfere with the scale evaluation (drugs or food poisoning, visual impairment or a low educational level).

### 2.3. Control Group

Using MoCA (Montreal Cognitive Assessment), we enrolled healthy people with MoCA assessment scores ≥26 and age, education level and gender matched with NMOSD group as the control group.

### 2.4. Clinical Assessment and Follow-Up

Demographic data, including gender, age and years of education were collected. Detailed medical history, physical examination and relevant clinical features were recorded. Routine laboratory tests, including thyroid function, were measured. Serum AQP-4 antibody titer was detected by a cytometric bead array (CBA) as described previously [19]. Follow-up (90 ± 7 days and 180 ± 7 days after enrollment) was conducted and serum AQP-4 antibody titer and MoCA were re-tested.

### 2.5. Assessment of Cognitive Function

Patients were assessed on six aspects of cognitive domain. The cognitive assessments were performed as follows: (1) Overall cognitive function assessment—MoCA [20] was used to assess overall cognitive function of all subjects. (2) Verbal learning and memory—the Chinese Auditory Language Learning Test (CAVLT) [21] was used to measure short-term and long-term auditory language memory. It consists of a 15-word list consisting of five learning tests, including interruption test, immediate recall test, delayed recall test and recognition test, with N1–N6 assessing immediate memory, N7 and N8 assessing long-term memory and N9 assessing cognitive memory. (3) Verbal fluency assessment—Subjects’ verbal fluency was assessed using the verbal fluency test (VFT) [22]. The scale requires the subjects to list as many words as possible in the category specified by the tester within one minimum, such as fruit and animals, being scored according to the correct number of words listed without repetition. (4) Assessment of information processing speed, working memory and attention—information processing speed, working memory and attention were assessed by the Paced Auditory Serial Addition Test version 3/2s (PASAT-3/2) [23]. In addition, the Stroop color-word test A (CWT-A) was used to assess visual search speed. The CWT-B was used to assess visual search speed with working memory and the Digit Span Test (DST) was used to assess attention [24]. (5) Evaluation of visual structure and visual-spatial memory ability—Rey–Osterrieth Complex Figure Test (ROCF) [25] was used to evaluate visual structure and visual-spatial memory. (6) Executive ability was evaluated by the CWT-C (Figure 1 and Table 1) [26].

### 2.6. Statistical Analysis

All data were rounded to two decimal places. The measurement data of normal distribution was expressed as mean ± standard deviation and the t-test was adopted to analyze the differences between acute NMOSD and the control group. Data that did not follow the normal distribution were described as quartiles and the rank-sum test was adopted to analyze. Enumeration data were described in terms of the number of cases and composition ratio as N (%). Spearman’s correlation analysis was adopted to analyze the correlation between multiple related factors (age, years of education, etc.) that may affect cognitive function in patients with acute NMOSD. Friedman’s ANOVA test was adopted to compare differences of cognitive function and serum AQP-4 antibody titer during follow-up. All statistical analyses were performed using IBM SPSS Statistics version 26.0 (IBM, New York, NY, USA) and the Statistical Analysis System 9.4 [27]. *p* < 0.05 was considered statistically significant.

## 3. Results

### 3.1. General Information of the Research Object

A total of 30 patients were recruited in this study. According to the criteria, 25 patients were actually enrolled (Two patients were blind on admission; two patients could not draw due to twitching of their upper limbs; one patient could not understand or speak Mandarin). The male to female ratio was 4:21. The average age was 44.80 ± 12.93 years and the average years of education was 9 (9, 12) years. In the control group, there were 30 subjects with a male to female ratio of 11:19. The average age was 47.27 ± 8.81 years, and the average years of education was 15 (12, 16) years. There was no significant difference in mean age between NMOSD and control groups in the acute phase (*p* = 0.423). There was no statistical difference in the years of education (*p* = 0.061).

#### 3.1.1. Comparison of Cognitive Evaluation Scales between NMOSD and Control

Significant differences were found in MoCA, CRAVLT-N7, CRAVLT-N8, ROCF-C, ROCF-R, PASAT-3, PASAT-2 and CWT-A between acute NMOSD and control group (*p* < 0.05), which were summarized in Table 2.

#### 3.1.2. Ratio of Cognitive Dysfunction in Acute NMOSD

Standard scores (Z-score) of each cognitive scale in NMOSD were calculated, respectively (Figure 2, Table 3). The number and proportion of acute NMOSD with cognitive impairment and cognitive domains were shown in Table 4. We found that 3 patients (12%) with a single cognitive domain dysfunction and 16 patients (64%) with two or more cognitive domain dysfunction were recorded. One patient had cognitive impairment in all five cognitive domains. 

#### 3.1.3. Changes of MoCA and Serum AQP-4 Antibody Titers during Follow-Up

MoCA score and AQP-4 antibody titers of NMOSD patients during follow-up visits were measured. We found significant differences in serum AQP-4 antibody titers during follow-ups (*p* < 0.001), while there was no statistical difference regarding the MoCA scores (Figure 3 and Figure 4). There was no statistical significance in serum AQP-4 antibody titer and MoCA scores (Figure 5, Table 5).

#### 3.1.4. Correlation Analysis of Cognitive Function and other Factors in Patients with NMOSD

During cognitive function assessment, the MoCA, CRAVLT-N6, PASAT-3, CWT-A and DST scores were positively correlated with years of education. The MoCA, ROCF-R and PASAT-2 scores were negatively correlated with intracranial lesions. The CWT-A scores were positively correlated with intracranial lesions (Table 6). And the result of the correlation between cognitive evaluation scores and related factors by Kendall and Spearman test were shown in Appendix A.

## 4. Discussion

Cognitive dysfunction can seriously affect patients’ quality of life and their prognosis. Currently, there are only a few studies investigating cognitive impairment in patients with NMOSD [14,17,28,29,30]. Potential mechanisms of cognitive impairment in NMOSD are speculated to be related to brain lesions and serum AQP-4-IgG positivity, while the evidence of potential correlation between serum AQP-4-IgG and cognitive impairment is lacking.

### 4.1. Summary & Contributions

In our study, we found that over 60% of acute NMOSD patients had cognitive impairment, which was reflected in MoCA scores adjusted for standard scores. A previous study indicated that there were 64% patients with two or more cognitive domain impairments [31]. Our results showed that cognitive dysfunction in Chinese patients with acute NMOSD is primarily manifested in terms of word memory, information processing speed, working memory and attention, visual structure, vasospastically memory and executive ability. Regarding word memory, NMOSD patients during the acute disease stage mostly had cognitive impairment in short-term memory and recognition memory. Consistent with this, He and colleagues indicated that NMOSD patients had cognitive domain impairments only in short-term word memory, while long-term and recognition memories were not impaired [12]. Another study inferred the opposite conclusion that NMOSD patients had cognitive impairment in both short and long-term memory, while the recognition memory remained normal [14]. We propose that the differences in sample size, populations, evaluating scales and criteria for cognitive impairment in NMOSD may explain some discrepancies. 

Cerebral edema was noted in patients with NMOSD during the acute phase, while the brain injury can shrink or disappear during the chronic phase [32]. Recently, it was proposed that inflammatory brain injury mediated by specific antibodies to NMOSD and the surrounding edema zone may damage the nerve fiber pathways or their corresponding nodes in the prefrontal cortex and other lobes, subsequently affecting cognitive function [8,9,33]. In clinical practice, we also observed cognitive impairment of NMOSD, and the serum AQP-4 antibody titer of patients gradually decreased with disease remission. However, we did not find that the MoCA score of patients had significant changes during the progress of the disease, and there was no significant correlation between the serum AQP-4 antibody titer and MoCA score.

Since the data in this study are continuous variables and do not conform to normal distribution, we used Spearman to analyze the correlation between the cognitive function scores of NMOSD patients and age, years of education and other factors. Consistent with previous studies [14,28], cognitive function of NMOSD patients is positively correlated with years of education. In fact, the simulated cube and bell chart tests in the MoCA are more difficult for people with less education. To avoid interference of supratentorial lesions with the cognitive outcome of this study, we excluded NMOSD patients with supratentorial lesions. We found that some of patients with subatentorial lesions eventually developed cognitive impairment. We believe that extensive cortical astrocyte damage or other specific biochemical reactions may play a role in cognitive impairment of NMOSD.

### 4.2. Benefits & Limitations

We used MoCA and other cognitive function scales to evaluate NMOSD patients in this study. The characteristics of cognitive dysfunction in Chinese NMOSD patients from various aspects were investigated, providing insights for the evaluation of cognitive function with respect to NMOSD in clinical practice. In this study, we also explored the correlation between the serum AQP-4 antibody titer in NMOSD patients and their cognitive function.

Our study has several limitations. First, the sample size is relatively small in this study since NMOSD is a rare disease. Further studies with a large number of patients are needed to verify our results. In addition, we did not include dynamic changes of brain MRI lesions of these NMOSD patients during follow-up.

### 4.3. Future Work

We plan to conduct subgroup analysis of AQP-4-positive and negative patients in future studies. Long-term follow-up is needed to reflect dynamic changes of cognitive function of NMOSD patients. Involving patients’ peripheral blood and cerebrospinal fluid to identify potential markers of cognitive dysfunction with respect to NMOSD would be of great interest.

## 5. Conclusions

In conclusion, our study suggests that more than 60% of Chinese patients with acute NMOSD have cognitive dysfunction, which can be affected by the years of education and the presence of intracranial lesions. In addition, dynamic changes of cognitive function in patients with NMSOD are not related to serum AQP-4 antibody titers.

## Figures and Tables

**Figure 1 brainsci-13-00090-f001:**
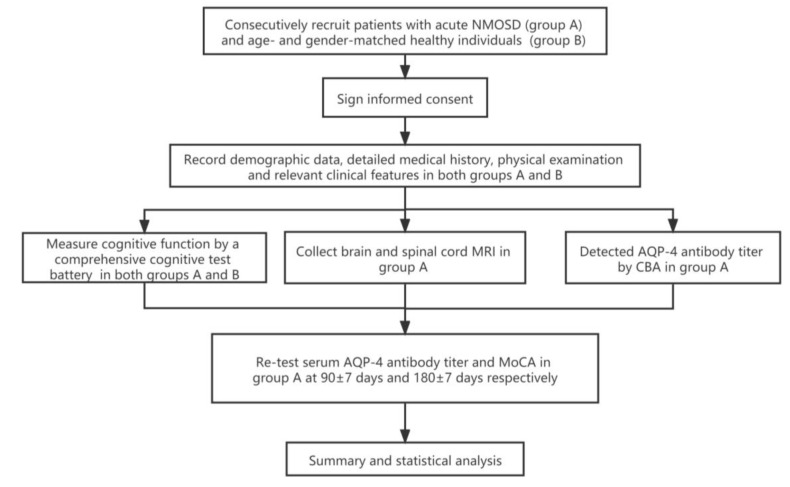
Trial profile. NMOSD: neuromyelitis optica spectrum disorders; AQP-4: aquaporin-4; CBA: cytometric bead array; MoCA: Montreal Cognitive Assessment.

**Figure 2 brainsci-13-00090-f002:**
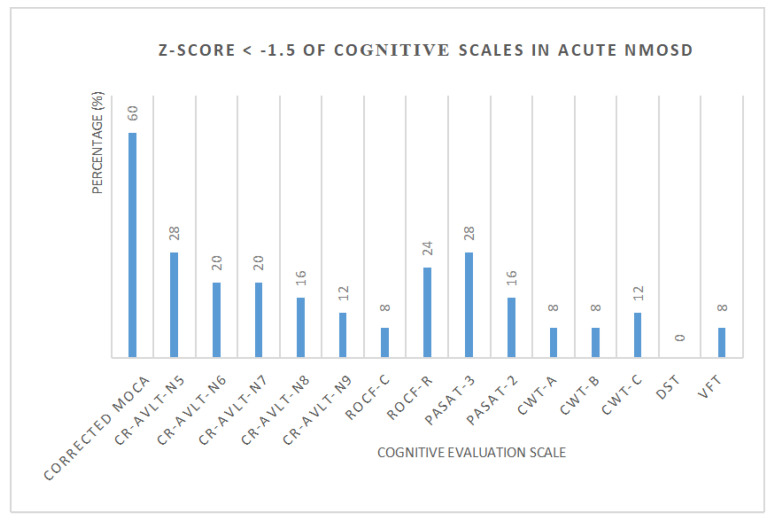
The percentage of acute NMOSD with cognitive impairment.

**Figure 3 brainsci-13-00090-f003:**
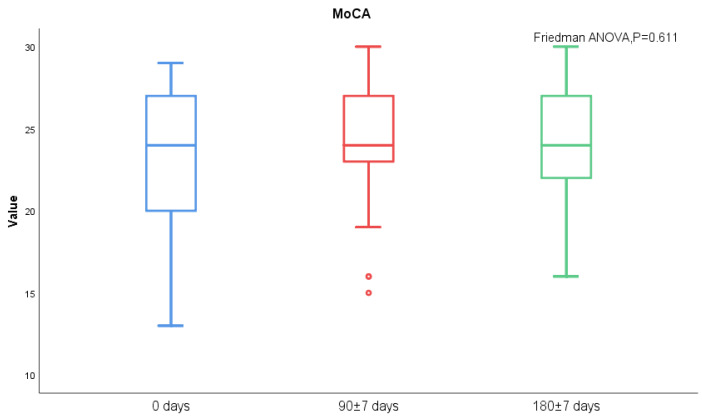
AQP-4 antibody titers between acute and chronic NMOSD.

**Figure 4 brainsci-13-00090-f004:**
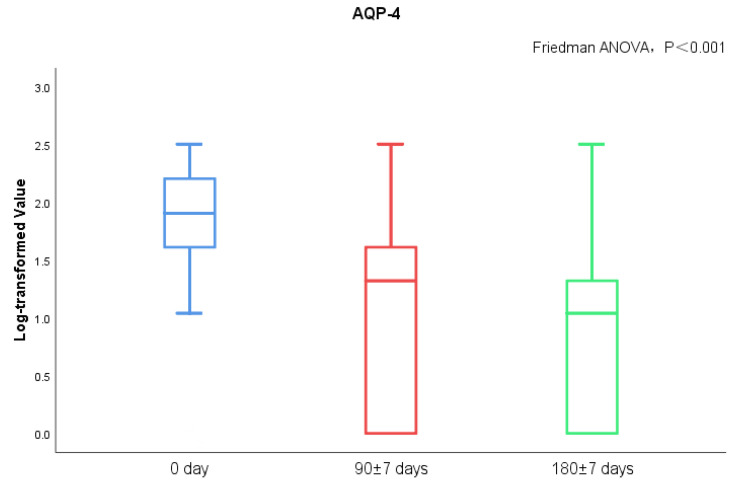
MoCA scores between acute and chronic NMOSD.

**Figure 5 brainsci-13-00090-f005:**
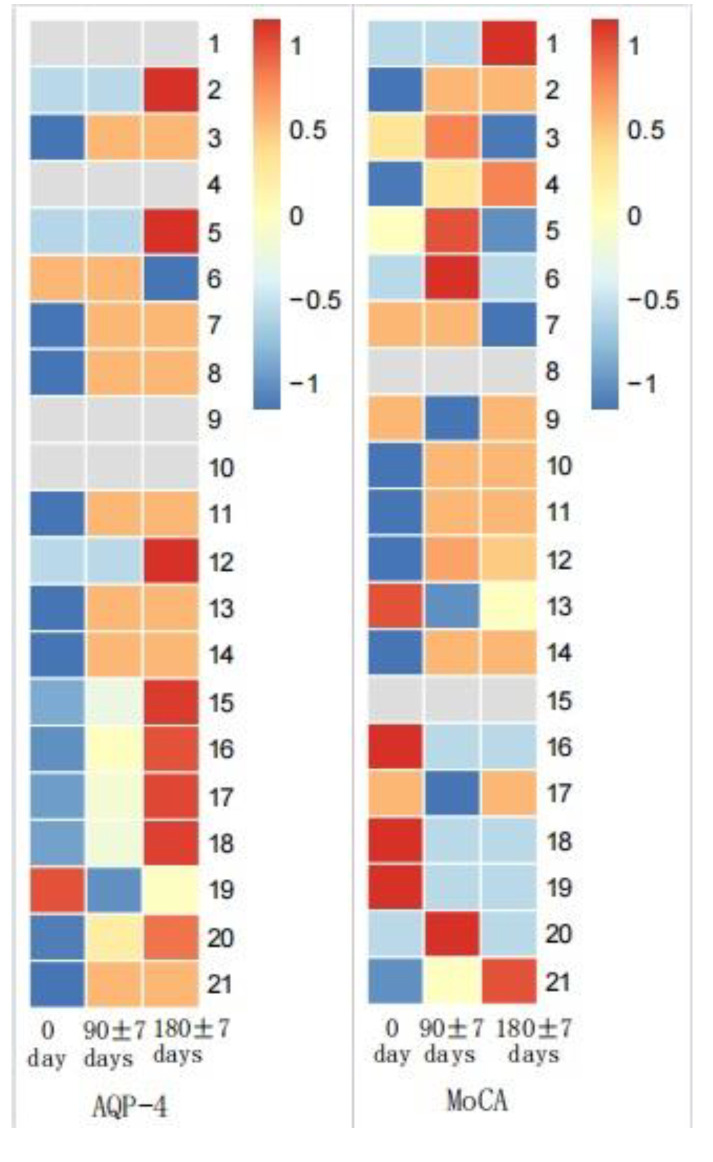
MoCA scores and AQP-4 antibody titers in NMOSD during follow-up.

**Table 1 brainsci-13-00090-t001:** Cognitive evaluation scales and their domains.

Cognitive Evaluation Scales	Domain
MoCA	Overall cognitive function
CRAVLT	Verbal learning and memory
ROCF	Visual structure and visual spatial memory ability
PASAT-3/2	Information processing speed, working memory and attention
CWT	Visual search speed, working memory, executive ability
DST	Attention
VFT	Verbal fluency

**Table 2 brainsci-13-00090-t002:** Comparison of cognitive evaluation scales between acute NMOSD and control group.

Cognitive Evaluation Scales (Score)	Acute NMOSD Group (n = 25)	Control Group (n = 30)	Statistical Magnitude	*p* Value
Corrected MoCA	24.00 (20.5, 27.5)	27.00 (26.75, 28.00)		<0.001 ***
CRAVLT-N1-5	37.96 ± 12.60	41.57 ± 7.21	t = 1.27	0.213
CRAVLT-N6	3.84 ± 1.68	4.37 ± 1.43	t = −1.26	0.213
CRAVLT-N7	7.72 ± 3.37	10.07 ± 2.41	t = −3.00	0.004 **
CRAVLT-N8	7.28 ± 3.48	9.47 ± 2.33	t = −2.68	0.011 *
CRAVLT-N9	14.00 (12.00, 15.00)	14.00 (13.00, 15.00)		0.072
ROCF-C	34.00 (31.75, 36.00)	36.00 (36.00, 36.00)		0.005 **
ROCF-R	11.92 ± 6.35	19.58 ± 4.87	t = −5.06	<0.001 ***
PASAT-3	35.00 ± 14.83	47.00 (36.00, 54.00)		0.013 *
PASAT-2	26.84 ± 12.59	34.00 (30.00, 42.50)		0.001 **
CWT-A (second)	28.67 ± 8.86	23.73 ± 4.59	t = 2.52	0.017 *
CWT-B (second)	38.36 ± 10.00	34.37 ± 8.11	t = 1.64	0.107
CWT-C (second)	67.69 (56.37, 82.79)	67.82 ± 13.23		0.800
DST	12.0 (11.00, 15.00)	14.10 ± 2.31		0.05
VFT	31.60 ± 12.26	36.47 ± 5.24	t = −1.85	0.074

* *p* < 0.05. ** *p* < 0.01. *** *p* < 0.001.

**Table 3 brainsci-13-00090-t003:** Z-Score < −1.5 of cognitive scales in acute NMOSD.

Cognitive Evaluation Scale	Z-Score < −1.5(The Number of People)	Percentage (%)
	(the Z-Score of CWT > 1.5)	
Corrected MoCA	15	60
CR-AVLT-N5	7	28
CRAVLT-N6	5	20
CRAVLT-N7	5	20
CRAVLT-N8	4	16
CRAVLT-N9	3	12
ROCF-C	2	8
ROCF-R	6	24
PASAT-3	7	28
PASAT-2	4	16
CWT-A	2	8
CWT-B	2	8
CWT-C	3	12
DST	0	0
VFT	2	8

**Table 4 brainsci-13-00090-t004:** Dysfunction of cognitive domains in acute NMOSD.

Cognitive Domain	Number of People with Disabilities	Percentage (%)
Verbal memory	13	52
Speech fluency area	2	8
Information processing, working memory, areas of attention	8	32
Visual spatial structure and visual spatial memory	9	36
Executive Capability Area	5	20

**Table 5 brainsci-13-00090-t005:** MoCA scores and AQP-4 antibody titers in NMOSD during follow-up.

	F	*p*
AQP-4 antibody titer (the second time)	0.127	0.728
AQP-4 antibody titer (the third time)	0.050	0.826
AQP-4 antibody titer (the first time): AQP-4 antibody titer (the third time)	4.381	0.057
AQP-4 antibody titer (the second time): AQP-4 antibody titer (the third time)	0.202	0.660

**Table 6 brainsci-13-00090-t006:** Correlation factor analysis.

Cognitive Evaluation Scale	Related Factors	R	*p*
MoCA	Years of Education	0.65	<0.001 ***
	Intracranial lesions	−0.569	0.007 **
CRAVLT-N6	Years of Education	0.638	0.002 **
ROCF-R	Intracranial lesions	−0.519	0.016 *
PASAT-3	Years of Education	0.477	0.029 *
PASAT-2	Intracranial lesions	−0.43	0.047 *
CWT-A	Years of Education	−0.463	0.035 *
	Intracranial lesions	0.518	0.016 *
DST	Years of Education	0.477	0.029 *

* *p* < 0.05. ** *p* < 0.01. *** *p* < 0.001.

## Data Availability

Not applicable.

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
