# Peer review of "The Characteristics of Cognitive Proficiency in Patients with Acute Neuromyelitis Optica Spectrum Disease and its Correlation with Serum Aquaporin-4 Antibody Titer"

_brainsci, 2023, doi:10.3390/brainsci13010090_

Round 1
Reviewer 1 Report
This research paper explored the characteristics and dynamic evolution of cognitive impairment in patients with neuro-myelitis optica spectrum disorder, which is a rare condition. This research topic is an interesting and important area of research. The paper is generally well written and merits publication; however, the quality of the paper can be enhanced if the following points can be addressed.
1. I suggest the authors to include a little more description about AQP4 and its correlation with Ccognitive functions in the introduction of the paper and cite related references.
2. The results are clear with proper statistical analysis.
3. The conclusion need to discuss about the potential benefits and shortcomings of this study and what steps are necessary in understanding and finding proper resolution for diagnosis in the patients suffering from this rare disease.
Author Response
- I suggest the authors to include a little more description about AQP4 and its correlation with cognitive functions in the introduction of the paper and cite related references.
Reply:Thank you for your suggestion. We have added a description of AQP-4 and its relationship with cognitive function in the introduction.(line 55-63)
- The results are clear with proper statistical analysis.
Reply:Thank you.
- The conclusion need to discuss about the potential benefits and shortcomings of this study and what steps are necessary in understanding and finding proper resolution for diagnosis in the patients suffering from this rare disease.
Reply:Thank you for your reminding. We have explained the significance and shortcomings of this study in “4.2. Benefits & Limitations” of the discussion.

Reviewer 2 Report
Dear Authors; I found this work an interesting exploration of the characteristics and dynamic evolution of cognitive impairment in patients with neu-33 romyelitis optica spectrum disorder. It needs some extra work prior to furthure processing. Regards. P.S.
[1] Writing:
1-1 References: Format them in MDPI standards. For example years are in bold.
1-2 Abbreviations: Add list of used abbreviation in the work right before the reference section.
1-3 Add numbers to the subsections of the each major section.
1-4 "4.Discussion". Add its subsections line this: "4.1.Summary & Contributions""4.2. Limitations", "4.3. Future Work", "4.4.Conclusion".
1-5 Add subsection "4.3. Future work" in one paragraph and suggest few potential directions.
1-6 Squeeze Table 1 in one page. It is hard to follow it. Same with other tables.
1-7 Section 3.1. needs title !
[2] Statistical:
2-1 Kendal correlation: You used Spearman correlation rather than Kendall. 2-1-1 Defend your choice in the discussion section. 2-1-2 Add the parallel results in the supplementary materials for readers comparison.
Link: https://en.wikipedia.org/wiki/Kendall_rank_correlation_coefficient
2-2 Missing statistical software: Add it in the "2.Materials & Methods" and cite its source in the reference section.
2-3 Matching Process between cases and controls: Explain it clearly in the "2.Materials & Methods" Was it propensity score ? Stratification, what ?
2-4 The study sample sizes are relatively low: 25 vs. 30. Report power of analysis in the work. Must be at least 80%.
Author Response
[1] Writing:
1-1 References: Format them in MDPI standards. For example years are in bold.
Reply:Thank you for your reminding. We have revised the format of references.
1-2 Abbreviations: Add list of used abbreviation in the work right before the reference section.
Reply:Thank you for your suggestion. We have added abbreviations before the references.
(line 327-331)
1-3 Add numbers to the subsections of each major section.
Reply:Thanks for your reminder, we have modified this.
1-4 "4.Discussion". Add its subsections line this: "4.1. Summary & Contributions""4.2. Limitations", "4.3. Future Work", "4.4.Conclusion".
Reply:Thank you for your suggestion. We have divided the discussion into several parts according to your requirements.
1-5 Add subsection "4.3. Future work" in one paragraph and suggest few potential directions.
Reply:Thank you. We have divided the discussion into several parts according to your requirements and added the future work part. (line 296- 304)
1-6 Squeeze Table 1 in one page. It is hard to follow it. Same with other tables.
Reply:Thank you for your reminding. We have modified the format of each table.
1-7 Section 3.1. needs title !
Reply:Thank you for your reminding. We have added the title of 3.1. (line 165)
[2] Statistical:
2-1 Kendal correlation: You used Spearman correlation rather than Kendall. 2-1-1 Defend your choice in the discussion section. 2-1-2 Add the parallel results in the supplementary materials for readers comparison.
Reply:Thank you for your question. Since the data in this study are continuous variables and do not conform to normal distribution, we use Spearman to analyze the correlation between the cognitive function scores of NMOSD patients and age, years of education and other factors.
Kendall correlation is used to reflect the correlation of categorical variables. It is applicable to the situation that both categorical variables are classified in an orderly manner. It is mainly used to evaluate the consistency of the results of the two detection methods. The data analyzed in this study are continuous variables, with the purpose of exploring the correlation between NMOSD patients' scores and their age, years of education and other factors. Therefore, Kendall correlation is not applicable to this study.
We also used Kendall correlation to analyze these data through SPSS software (the results are put in the supplementary materials)
Link: https://en.wikipedia.org/wiki/Kendall_rank_correlation_coefficient
2-2 Missing statistical software: Add it in the "2.Materials & Methods" and cite its source in the reference section.
Reply:Thank you for your reminding. We have added the statistical software in the "Materials & Methods". (line 161-162)
2-3 Matching Process between cases and controls: Explain it clearly in the "2.Materials & Methods" Was it propensity score ? Stratification, what?
Reply:Thank you for your reminding. We have added a description of the inclusion criteria for the control group. (line 98-101)
2-4 The study sample sizes are relatively low: 25 vs. 30. Report power of analysis in the work. Must be at least 80%.
Reply:Thank you for your question. The research group used the sample size calculator(Link: https://clincalc.com/stats/samplesize.aspx) to estimate the enrolled samples before the start of the study. It is estimated that the incidence of cognitive dysfunction in the NMOSD group is 26%, the incidence of cognitive dysfunction in the control group is 0%, and the ratio between the NMOSD group and the control group is 1:1. The calculation results suggest that the sample size of the study is at least 25 people in each group. Influenced by the epidemic situation and the low incidence rate of NMOSD, only 25 cases of NMOSD and 30 controls were successfully included in this study.

Round 2
Reviewer 2 Report
Dear Authors; my main concerns were addressed satisfactorily. Regards.